# The Role of Zinc and NMDA Receptors in Autism Spectrum Disorders

**DOI:** 10.3390/ph16010001

**Published:** 2022-12-20

**Authors:** Kevin Lee, Zoe Mills, Pangying Cheung, Juliette E. Cheyne, Johanna M. Montgomery

**Affiliations:** Department of Physiology and Centre for Brain Research, University of Auckland, Auckland 1010, New Zealand

**Keywords:** NMDA receptor, synaptic plasticity, autism spectrum disorder, glutamate, zinc

## Abstract

NMDA-type glutamate receptors are critical for synaptic plasticity in the central nervous system. Their unique properties and age-dependent arrangement of subunit types underpin their role as a coincidence detector of pre- and postsynaptic activity during brain development and maturation. NMDAR function is highly modulated by zinc, which is co-released with glutamate and concentrates in postsynaptic spines. Both NMDARs and zinc have been strongly linked to autism spectrum disorders (ASDs), suggesting that NMDARs are an important player in the beneficial effects observed with zinc in both animal models and children with ASDs. Significant evidence is emerging that these beneficial effects occur via zinc-dependent regulation of SHANK proteins, which form the backbone of the postsynaptic density. For example, dietary zinc supplementation enhances SHANK2 or SHANK3 synaptic recruitment and rescues NMDAR deficits and hypofunction in *Shank3^ex13–16−/−^* and *Tbr1^+/−^* ASD mice. Across multiple studies, synaptic changes occur in parallel with a reversal of ASD-associated behaviours, highlighting the zinc-dependent regulation of NMDARs and glutamatergic synapses as therapeutic targets for severe forms of ASDs, either pre- or postnatally. The data from rodent models set a strong foundation for future translational studies in human cells and people affected by ASDs.

## 1. Introduction

Glutamate receptors are the principal mediators of excitatory neurotransmission at central nervous system synapses. The two major subtypes of glutamate receptors that cluster at the postsynaptic membrane are N-methyl-d-aspartate (NMDA) and a-amino-3-hydroxy-5-methylisoxazole-4-proprionic acid (AMPA) receptors. NMDA glutamate receptor subtype is crucial for synapse development, neuroplasticity, and pathological neurotoxicity, where its level at the synapse critically regulates brain function [1,2].

Distinct from other types of ligand-gated ionotropic receptors, NMDARs exhibit several unique biophysical and functional properties critical to their key role in the cellular mechanisms thought to underlie the initiation of changes in synaptic strength, and the formation of new neural networks [3]. Firstly, NMDARs are highly permeable to calcium [4,5] and require allosteric co-agonist binding of glycine, or D-serine, in addition to glutamate to open [6,7,8]. Furthermore, the existence of a voltage-dependent block by Mg^2+^ requires the binding of agonists to be concurrent with sufficient depolarization of the postsynaptic membrane in order to open the NMDAR channels [9]. As such, NMDAR acts as a ‘coincidence detector’ to initiate changes in synaptic strength that lead to the formation of new neural networks and cellular mechanisms that are thought to underlie learning and memory [3]. In comparison to AMPARs, NMDARs also exhibit slow kinetics due to slow glutamate unbinding [10,11] and facilitate a diverse range of interactions, owing to several extracellular modulatory sites and long carboxyl-terminal domains (CTDs) that extend intracellularly.

### 1.1. NMDAR Subunit Composition and Functional Consequences

NMDARs are assembled as heteromeric tetramers that differ in subunit composition, depending on brain region, cell type, subcellular localisation, and developmental stage. There are seven different subunits, which are classified into three subfamilies according to sequence homology: the GluN1 subunit; four distinct GluN2 subunits (GluN2A, GluN2B, GluN2C, and GluN2D) which are encoded by four different genes; and two GluN3 subunits (GluN3A and GluN3B) which are encoded by two separate genes [12,13,14,15]. Functional NMDARs are thought to be composed of two obligatory GluN1 subunits and two copies of GluN2 subunits and/or GluN3 subunits [16,17]. The existence of eight splice variants in the GluN1 subunit subfamily further adds a multiplicity of receptor subtype compositions in the brain. The critical channel properties of NMDARs, such as Mg^2+^ blockade, Ca^2+^ permeability, and single-channel conductance, are all largely controlled by a single GluN2 subunit residue and how it interacts with the GluN1 subunit [18].

In contrast to the GluN1 subunit, which is ubiquitously expressed from the prenatal period to adulthood in rodent brains, a heterogeneous spatiotemporal expression pattern has been observed for the GluN2 subunits [19]. GluN2B and GluN2D are expressed in the embryonic brain and spinal cord, predominantly concentrated in the cortex and thalamus, or midbrain, respectively, but soon after birth, GluN2A expression increases and it becomes abundantly expressed throughout the CNS by adulthood [20,21,22]. GluN2D expression, however, decreases dramatically after birth, whereas GluN2B expression is maintained at high levels in the cortex and hippocampus, and at moderate levels in the midline structures following birth [20,21,22]. Subsequently, the expression of GluN2A increases and is abundant in both the hippocampal dentate gyrus and throughout the cortex by adulthood [20,21,22]. GluN2C expression initiates late in development and is mainly confined to the olfactory bulb and the cerebellum [21,22]. On the other hand, the GluN3A subunit expression peaks in early postnatal days in the forebrain, cortex, hippocampus, thalamus, and spinal cord, and then declines progressively, while GluN3B expression rises slowly throughout development and is primarily located in the regions of the neocortex, hippocampus, striatum, cerebellum, brainstem, and spinal cord [23,24]. The age-dependent arrangement of subunit types is thought to mediate altered kinetics and binding properties underlying changes in NMDAR control of neural plasticity during development and upon maturation [25]. Thus, it is hypothesised that the GluN2D and GluN3A subunits play crucial roles early in development, such as synaptogenesis and synapse maturation, while GluN2A and GluN2B, the predominant subunits in the adult CNS, may have important roles in synaptic function and plasticity.

Subcellularly, di-heteromeric GluN1/GluN2A and tri-heteromeric GluN1/GluN2A/GluN2B are the predominant NMDARs located at the synapses of the adult forebrain. In contrast, GluN2B-containing NMDARs are prevailing at peri- and extra synaptic sites [26,27]. However, classifying subcellular localization—synaptic versus extrasynaptic—of NMDARs based on the expression of GluN2A or GluN2B is an over-simplification [28,29]. In the hippocampus, NMDARs have been shown to be mobile and can rapidly exchange between synaptic and extrasynaptic compartments through lateral membrane diffusion, further increasing the heterogeneity of NMDAR subtypes at both synaptic and extrasynaptic sites throughout the brain [30,31].

Despite sharing over 70% of sequence homology, the GluN2A and GluN2B subunits have several distinct differences [16]. These differences include GluN2A containing higher NMDAR channel open probability and significantly faster deactivation kinetics than GluN2B [32,33], in addition to altered glutamate sensitivity and agonist affinity [34]. However, the notable divergence in amino acid sequence is attributed to the regions encoding the CTDs responsible for the profile of intracellular molecules in which the subunit can interact with and, thus, the downstream signalling pathways the subunit mobilises. The developmental change in the ratio of the GluN2A and GluN2B subunits has been proposed to result from several phosphorylation events of the CTDs, affecting the stability of the subunits within the NMDAR formation and altering the binding affinity of the motifs which interact with endocytic machinery, ultimately leading to the diminished presence of specific subtypes [35,36].

### 1.2. NMDARs and Synaptic Plasticity

In response to the strong afferent activity-induced depolarization of postsynapses, which is coincident with presynaptic transmitter release, calcium influx through the NMDARs triggers the active insertion or removal of AMPARs [37]. These receptor dynamics underlie the major forms of synaptic plasticity where increases in synaptic strength are termed long-term potentiation (LTP) [38] while decreases in synaptic strength are termed long-term depression (LTD) [39]. Thus, AMPARs are thought to be a major mechanism responsible for the expression of synaptic plasticity, while NMDARs for their control.

During the expression of LTP, the insertion of AMPARs has largely been shown to not be accompanied by a parallel insertion of NMDARs [40,41,42]. This observation has led to the hypothesis that NMDARs are not subjected to rapid recycling into and out of the synaptic membrane as seen with AMPARs [43,44,45]. This can be dependent, however, on the method of LTP induction and the site of LTP expression (i.e., pre- or postsynaptic) [46,47]. NMDAR-mediated responses can be both up- and down-regulated. For example, pharmacological activation of G-coupled receptors, such as metabotropic glutamate receptors (mGluRs) or muscarinic acetylcholine receptors, has been shown to potentiate NMDAR-mediated currents via protein kinase C (PKC) activation [48,49]. In neurons, pharmacological activation of PKC increases NMDAR channel open time and potentiates NMDAR-mediated currents via an increase in exocytic NMDAR delivery to the postsynaptic membrane [50,51,52]. More recently, rapid recycling of NMDARs and enhancement of their surface expression have been shown to occur with synaptic potentiation through activity-dependent phosphorylation of GluN2A [53]. NMDAR-mediated currents can also be regulated by synaptic stimulation, particularly in the negative direction. The induction of LTD suppresses NMDAR-mediated currents [42,54,55,56]. Evidence of NMDAR endocytosis following the application of exogenous agonists has been shown in heterologous expression systems and in neurons [57,58,59]. Both NR2A and NR2B subunits contain endocytic motifs in their C-termini [35,48]. These motifs have different affinities for binding the endocytic machinery, which could translate to a subunit-regulated ability to undergo endocytosis [35] following the induction of LTD [60].

Activity-dependent regulation of NMDARs is dependent on the state of the synapse [54,60]. Electrophysiology has shown that synapses exist in distinct states, and, with the induction of synaptic plasticity, synapses move between these states [60]. To date, synaptic states include silent, recently silent, active, potentiated, and depressed, with each state being defined by the ability of both AMPA and NMDA receptors to undergo activity-dependent regulation via LTP or LTD [42,53,60]. What state a given synapse is in and what state it has recently occupied then determine the ability of the synapse to undergo future synaptic plasticity and what mechanisms it employs to do so. These data indicate that NMDARs are not as static in the postsynaptic membrane as previously thought, but may in fact be as dynamic as AMPARs during synaptic plasticity, with several higher-order control mechanisms acting upon the receptors.

### 1.3. Zinc and NMDARs

Zinc is an endogenous divalent cation known to be a potent modulator of NMDARs and other receptor subtypes at synapses (Figure 1). Stored in glutamatergic synaptic vesicles, zinc accumulates in the presynaptic terminal and is co-released with glutamate [61,62,63,64,65]. Chelatable or free zinc within glutamatergic synaptic vesicles is at the millimolar range, where it is accumulated by a vesicular zinc transporter, ZnT3 [66,67,68]. At rest, the ambient extracellular synaptic zinc levels are remarkably low (<10 nM), even at the mossy fiber-CA3 synapses in the hippocampus which are the most zinc-abundant synapses in the brain [69]. When co-released with glutamate, the concentration of zinc increases transiently to reach a sub-micromolar range that can inhibit GluN2A-subunit-containing NMDARs [69] (Figure 1). The inhibition of NMDARs at this concentration occurs independently of membrane voltage through the binding of zinc at the high-affinity site in the N-terminal domain of the GluN2A subunit [70,71,72]. The homologous N-terminal domain of the GluN2B subunit can also be bound by zinc to inhibit GluN2B-containing NMDARs, but at a lower affinity (IC_50_ ∼2 μM) [70,73,74,75], and both GluN2A- and GluN2B-containing NMDARs can be blocked by zinc in a voltage-dependent manner in the micromolar range (IC_50_ ≅ 20 μM at −40 mV) [71,76]. While free zinc can be transported into organelles, such as mitochondria, endoplasmic reticulum, and the Golgi apparatus, and stored [77], the majority of zinc is bound by synaptic proteins.

Vesicular zinc release is required for the induction of presynaptic mossy fiber LTP by changing the presynaptic transmitter release probability [78,79,80] (Pr), and it has also been shown to enhance postsynaptic LTP in area CA1 of the hippocampus in a concentration-dependent manner via the subunit-specific modulation of NMDAR and P2X receptors [81,82,83,84]. Zinc can enter the postsynaptic neuron via NMDARs, but also via other channels, including calcium-permeable AMPARs, voltage-gated calcium channels, and TRPM7 channels [34,85,86,87,88], where it becomes highly enriched within dendritic spines [77,89] and modulates synaptic transmission and plasticity [63,80,81,82,83] through mechanisms which are poorly understood. Once in the spine, buffers, such as metallothionein III, likely facilitate the sequestration of free zinc [90,91,92]. Recent work has demonstrated that zinc can also regulate glutamatergic synapses through the recruitment or alteration of postsynaptic density (PSD) proteins. A key example is the SHANK family of proteins (SHANK1, SHANK2, and SHANK3), which are localised at the core of the postsynaptic density where they have been shown to modulate the structure, plasticity, and maturation of synapses [93,94,95,96]. As such, they are often described as the “master regulators” of synapses as they bind to many proteins, receptors, and the actin cytoskeleton to form the focal point of the synaptic scaffold. Specifically, SHANK proteins are characterised by an extensive set of protein-to-protein interaction domains: ankyrin repeats, Src homology 3 (SH3) domain, PSD95/DlgA/Zo-1 (PDZ) domain, a proline-rich/homer and contactin binding domain, and a C-terminal sterile alpha motif (SAM) domain [93,94]. In particular, the SH3 domain of SHANK has been reported to interact with a glutamate-receptor-interacting protein (GRIP), which is involved in AMPAR endocytosis and synaptic plasticity via interaction with the GluA2 subunit [97,98,99]. On the other hand, the PDZ domain of Shank interacts with SAP90/PSD95-associated protein (SAPAP; also known as guanylate kinase-associated protein [GKAP]), which provides a link to NMDARs through PSD95 [100]. This, in turn, provides a bridge between the NMDAR-/AMPAR-Stargazin-PSD95 complex with group I mGluRs and Homer at excitatory glutamatergic synapses, and contributes to synaptogenesis, synaptic transmission, and plasticity [101,102,103]. SHANK2 and SHANK3 are highly regulated by zinc, and specifically, zinc enhances SHANK2/3 stability and localisation at synapses [89,104] (Figure 2). Zinc is highly enriched in the postsynaptic density, where it is thought to bind to high-affinity zinc binding sites at the C- terminal SAM domains of SHANK2/3 [104]. Altering zinc levels changes the configuration of the synaptic scaffold and the localisation of SHANK2/3: In the presence of zinc, synaptic helical polymer “sheets” of self-associated SHANK2/3 scaffolding form in the postsynaptic density and increase the synaptic density [89,104,105]. In contrast, zinc depletion induces the disintegration of the postsynaptic density, likely due to a decrease in rigid subcellular architecture provided by these SHANK2/3 sheets. These data suggest that there is a concerted action of zinc and SHANK2/3 to increase the structural integrity and synaptic plasticity machinery of the glutamatergic postsynaptic density.

An important unresolved area that requires further research is the role that SHANKs and zinc play in glutamatergic synaptic transmission, especially with regard to NMDARs and subsequent plasticity. Studies of synaptic plasticity at CA1 synapses in the hippocampus have found that zinc has a profound effect on the induction of LTP, but not its maintenance [81,82,83]. Moreover, synapses lacking SHANK3 fail to undergo hippocampal LTP [106,107,108,109], suggesting that zinc and SHANK3 may operate in concert to regulate this facet of synaptic plasticity. Acute zinc alone may be insufficient to induce long-lasting forms of plasticity, as the AMPAR-mediated currents of hippocampal cultured neurons have been found to return to baseline levels within minutes after its removal [105]. A likely partner is the co-activation of NMDARs, which, as described above, is known to be critical for the induction of LTP [110,111]. Zinc-dependent activation of SHANK3 could, therefore, promote increased NMDAR and/or AMPAR recruitment to synapses, similar to what is known to occur with actin polymerization [45,112,113] and expression of LTP [114,115,116,117,118,119].

## 2. Pathological Effects of ASDs on NMDARs

Autism spectrum disorders (ASDs) are clinically diagnosed by persistent deficits in behavioural symptoms in two core criteria: (1) restrictive, repetitive behaviours, and (2) social communication and interaction. The American Psychiatric Association’s *Diagnostic and Statistical Manual, 5th Edition* (DSM-5) describes persistent deficits must occur in three areas of social communication and interaction, plus there must be at least two of four types of restricted, repetitive behaviours (Diagnostic and Statistical Manual of Mental Disorders: DSM-5™) [120]. Numerous known causative mutations have been identified in people affected by ASDs, and, when these genetic mutations are functionally grouped, many ASD genetic mutations converge on specific biological pathways involved in glutamatergic synapse function and plasticity.

Given their important roles in synaptic transmission and plasticity, as well as learning and memory, it comes as no surprise that disrupted NMDAR function has been implicated in various psychiatric disorders, including ASDs [121]. Multiple genetic variants of the NMDAR gene family (*GRIN*) have been detected in people with ASDs [122,123]. These include missense and frame-shift mutations and splice variants found throughout the entire *GRIN* gene, inclusive of the amino-terminal domain, transmembrane domains, linker regions, and carboxy-terminal domain [124]. So far, ASD-associated mutations and rare variants have been detected in all NMDAR subunit genes, including *GRIN1* encoding GluN1 subunit [118,119,120,121], *GRIN2A* encoding GluN2A subunit, *GRIN2B* encoding GluN2B subunit [125,126,127,128,129,130,131,132,133,134,135,136], *GRIN2C* encoding GluN2C subunit, and *GRIN2D* encoding GluN2D [125,126,127,128,129,130,131,132,133]. However, of all human NMDAR subunit genes implicated in ASDs, *GRIN2B* is the most predominant and recurrent ASD-risk gene [124]. Functional analyses that investigated the impact of ASD-associated *GRIN2B* mutations on the pharmacological and biophysical properties of NMDARs have revealed alterations in glutamate potency, receptor desensitisation, probability of channel opening, Mg^2+^ and Ca^2+^ permeability, charge transfer, receptor trafficking and surface expression, dendritic growth and spine density, and synaptic transmission [124,134,135,136,137,138,139,140,141,142,143,144,145,146,147,148]. Mice constitutively expressing 5–10% of normal GluN1 subunit (i.e., NR1^neo−/−^ mice) demonstrated autistic behaviours, including reduced social interactions, impaired pre-mating ultrasonic vocalisations, increased repetitive behaviours, and self-injury [149]. Moreover, mice that were haploinsufficient with an autistic patient-derived GluN2B-C456Y mutation (i.e., *Grin2b^+/C456Y^* mice) showed reduced GluN2B and GluN1 protein levels and impaired GluN2B-containing NMDAR currents at the Schaffer collateral-CA1 pyramidal (SC-CA1) synapses in the hippocampus [150]. Behaviourally, *Grin2b^+/C456Y^* mice displayed hypoactivity, anxiolytic-like behaviour, and moderately enhanced self-grooming in adulthood, although social interaction and communication were normal [150]. Together, increasing evidence suggests that perturbed NMDAR signalling may be the core molecular mechanism behind the pathogenesis of idiopathic autism.

### 2.1. Role of SHANKs with NMDARs in ASDs

Other than ASD-associated mutations directly linked to the NMDAR, many ASD risk genes involved in the structural and functional integrity of synapses have been discovered [151,152], in particular the *SHANK* gene family [153,154,155,156]. All three *SHANK* genes (*SHANK1*, *SHANK2,* and *SHANK3*) are implicated in ASDs, but autistic people with *SHANK3* mutations display more severe behavioural deficits [156]. Animal models targeting ASD-associated *Shank* mutations exhibit ASD-associated behavioural phenotypes, including impaired sociability and communication, enhanced repetitive self-grooming, and anxiety [157,158,159,160,161,162,163,164,165,166,167,168,169,170,171,172,173,174,175,176,177,178,179,180,181,182,183,184,185,186,187,188,189,190,191,192,193,194,195]. A summary of shankopathies in the developing brain is provided in Table 1 in our recent review [195]. Interestingly, many of these animal models demonstrated an alteration in the protein levels of NMDAR subunits, in NMDAR-mediated synaptic transmission, or in synaptic plasticity, such as LTP or LTD [106,113,158,161,163,165,166,167,168,170,175,182,185,187,188,192,194,195,196,197], highlighting the close functional coupling between SHANKs and NMDARs in ASDs.

Although NMDAR dysfunction is a common phenotype in ASD-associated *Shank* animal models, intriguingly, the details of the deficits in NMDAR signalling differ between ASD animal models, dependent on the type of *Shank* isoforms or specific domain targeted within *Shank*s, animal species, brain region, or cell type studied [198]. Mice that were haploinsufficient or deficient of the ankyrin repeat (ANK) domain of SHANK3, *Shank3^e4–9+/−^*, or *Shank3^e4–9−/−^*, respectively, demonstrated a significant reduction in NMDA/AMPA excitatory postsynaptic current (EPSC) ratio at cortical excitatory synapses onto striatal medium spiny neurons [168], and reduced hippocampal LTP at SC-CA1 synapses [106,167,196]. Similarly, rats with mutations targeting the ankyrin repeat domain (exon 6) also displayed impaired NMDAR-dependent, high-frequency, and stimulation-induced LTP at hippocampal SC-CA1 synapses [165]. Proteomic analysis of postsynaptic density (PSD) fractions from the striatum of *Shank3^e11−/−^* mutant mice (targeting Src homology 3 (SH3) domain) showed a decrease in GluN1 and GluN2B protein levels [199]. In contrast, NMDAR-mediated synaptic transmission at striatal synapses was normal in mice with a deletion of PSD95/DlgA/Zo-1 (PDZ) domain-coding exons 13–16 in *Shank3* (*Shank3^13–16−/−^*) [176], but the decay kinetics of NMDAR-mediated EPSCs were altered in *Shank3^13–16−/−^* mice [161], suggestive of a change in NMDAR subunit composition. *Shank3^e21−/−^* mice, deficient in a proline-rich domain, not only demonstrated a decrease in NMDA/AMPA ratio and reduced LTP in the CA1 region of the hippocampus, but also showed significantly diminished NMDAR-mediated synaptic responses and synaptic protein levels of GluN1 and GluN2A subunits in the prefrontal cortex [113]. Instead of targeting specific domains within the *Shank3* protein, mice with a complete *Shank3* knockout via deleting exons 4–22 (*Shank3^e4–22−/−^*) also showed a reduced synaptic GluN2A subunit protein level in the hippocampus (but not in the striatum) [185]. In comparison, a rat model with a complete knockout (a deletion spanning exons 11–21) displayed normal GluN1 and GluN2A protein levels in hippocampal and striatal PSD fractions and unaltered NMDAR-mediated EPSCs at SC-CA1 synapses, but theta burst stimulation-induced hippocampal LTP was reduced [200]. Similar to *Shank3* knockout mice, mice with human autism mutations also exhibited NMDAR dysfunction. These included impaired hippocampal LTP and decreased NMDA/AMPA ratio at corticostriatal synapses in mice with a transcriptional stop cassette inserted upstream of the PDZ domain-coding exon 13 in *Shank3* (*Shank3^+/E13^* and *Shank^−/E13^*) [167]; a decrease in NMDA/AMPA ratio at hippocampal SC-CA1 synapses in *Shank3^InsG3728+/+^* mice (insertion of a single guanine nucleotide located at position 3728, resulting in a frameshift that causes a premature truncation); and significantly reduced NMDAR-mediated EPSCs at corticostriatal synapses in *Shank3^InsG3680+/+^* (single guanine nucleotide insertion at position 3680 which led to a frameshift and a downstream stop codon) [192].

Comparable to *Shank3* mutant animals, rodent models with mutation targeting *Shank2* also showed alterations in NMDAR function. Two *Shank2* mutant mice, with a deletion in exons 6–7 or exon 24 (*Shank2^e6−7−/−^* and *Shank2^e24−/−^*, respectively), demonstrated a reduced NMDA/AMPA ratio at hippocampal SC-CA1 synapses [175,187]. Both hippocampal LTP and LTD were severely perturbed in *Shank2^e6−7−/−^* mice, possibly contributing to the spatial learning and memory deficits observed in these mice [187]. Furthermore, *Shank2^e6−7−/−^* mice exhibited impaired motor coordination in association with reduced levels of excitatory postsynaptic membrane proteins, including the NMDAR subunit GluN2C in the cerebellum [163]. Intriguingly, another line of *Shank2* mutant mice, which targeted the PDZ domain of SHANK2 but with a deletion of only exon 7 (i.e., *Shank2^e7−/−^*), demonstrated NMDAR hyperfunction rather than hypofunction as identified in *Shank2^e6−7−/−^* mice, although both *Shank2^e6−7−/−^* and *Shank2^e7−/−^* mice displayed comparable autistic-like behavioural traits, such as repetitive grooming and abnormal social behaviours [180,187]. *Shank2^e7−/−^* mice showed upregulated NMDAR subunit expression in the hippocampus, increased NMDA/AMPA ratio, and enhanced NMDAR-dependent hippocampal LTP [180]. Similarly, *Shank2* knockout rats also demonstrated enhanced NMDAR-mediated EPSCs, although LTP and LTD were impaired in the hippocampus [201]. Together, these findings provide important information that autistic behaviours contributed, in part, by the disruption in NMDAR signalling can be bidirectional.

The heterogeneity in NMDAR function induced by distinct ASD-associated *Shank2* or *Shank3* mutations in animal models is expected, as SHANKs present discrete spatiotemporal expression patterns [89,202,203] and interact specifically with synaptic protein isoforms [204,205]. Moreover, the neurophysiological effect of *Shank* mutations is dependent on neuronal cell types at different synapses [158,163,172,177,195,206]. Another crucial factor to consider is the age-dependent discrepancy in ASD-associated *Shank* mutation-induced NMDAR function that varies between different *Shank2* or *Shank3* mutant animals [195,207]. For example, *Shank2^e6−7−/−^* demonstrated NMDAR hyperfunction at the hippocampal SC-CA1 synapses at postnatal day 14 (P14), which switched to hypofunction when beyond P21 [208]. However, *Shank2^e7−/−^* and *Shank2^e24−/−^* mice did not show such a developmental switch, and for *Shank2^e7−/−^* mice, the increased NMDA/AMPA ratio was observed throughout development [197]. Moreover, *Shank3^e13−16−/−^* mice showed enhanced synaptic cortical hyperactivity in striatal spiny projection neurons during the second and third postnatal weeks [209,210]. NMDAR-mediated EPSC amplitude was observed to be normal at corticostriatal excitatory synapses in *Shank3^e13−16−/−^* mice at three weeks of age, but then NMDAR hypofunction was observed at 9–10 weeks of age [211]. Together, these data suggest that we need to consider the precise temporal mechanisms occurring with NMDAR function in ASDs.

Unlike *Shank2* and *Shank3* mutant mice, no NMDAR-specific malfunction was observed in *Shank1* mutant mice, although these animals displayed behavioural deficits, such as increased anxiety-related behaviours; impaired long-term memory retention and contextual fear memory; reduced motor function; and decreased levels of ultrasonic vocalizations and scent marking [166,181,186].

### 2.2. SHANK-Independent Regulation of NMDARs in ASDs

In addition to *SHANKs*, ASD-associated mutations in many other genes that encode synaptic proteins, that are localised at excitatory glutamatergic synapses, have also been found to alter NMDAR function. This includes insulin receptor tyrosine kinase substrate of 53 kDa (IRSp53), also known as brain-specific angiogenesis inhibitor 1-associated protein 2 (BAIAP2), which is a PSD component at excitatory synapses that interacts with PSD95 and *SHANKs* and regulates actin polymerisation, thereby modulating spine development and plasticity [212]. A single-nucleotide polymorphism variant and *de novo* copy number variants located at *BAIAP2* have been discovered in autistic individuals [213,214,215]. In addition, *IRSp53^−/−^* mice showed impaired social interaction and social communication [216]. At the SC-CA1 synapses in the hippocampus, NMDAR-mediated excitatory synaptic responses were enhanced and LTP was profoundly enhanced in *IRSp53^−/−^* mice [216,217], whereas NMDAR function was normal at the cortical excitatory synapses of layer 2/3 pyramidal neurons in the medial prefrontal cortex [216]. Another postsynaptic protein is synaptic adhesion-like molecule 1 (SALM1; also known as leucine-rich repeat and fibronectin type III domain containing, LRFN2), which interacts with PSD95 to modulate NMDAR clustering at synapses [218]. Genetic alterations in *LRFN2* have been implicated in ASDs, and mutant mice lacking SALM1 (*Lrfn2^−/−^* mice) displayed suppressed ultrasonic vocalization and increased acoustic startle, although learning and memory, social interaction, and repetitive behaviours were normal [219]. Interestingly, NMDAR-mediated synaptic transmission, as measured by NMDA/AMPA ratio and NMDA input–output ratio, was enhanced in the hippocampal CA1 region of *Lrfn2^−/−^* mice, while NMDAR-dependent LTP and LTD were suppressed [219]. As suggested by the authors [219], such a discrepancy (i.e., enhanced NMDAR-mediated synaptic response vs. reduced NMDAR-dependent synaptic plasticity) could be a result of a compensatory mechanism induced by erroneous synaptic plasticity and/or unexpected NMDAR hyperfunction causing changes in the molecular pathways downstream of NMDAR activation, which might have altered synaptic plasticity. This study signifies the complexity of molecular mechanisms behind defective NMDAR function in ASDs.

NMDAR dysfunction has also been observed in functional analyses of ASD-associated alterations in genes that encode molecules involved in the synaptic cell-adhesion pathways [220]. An example is Netrin-G ligand 2 (NGL-2)/LRRC4 (leucine-rich repeat containing 4), a postsynaptic adhesion molecule that interacts with PSD-95, an abundant excitatory postsynaptic scaffolding protein [221], and trans-synaptically with netrin-G2, a presynaptically expressed adhesion molecule [222]. Genetic variations in *LRRC4* in autistic individuals are well reported [223,224,225]. In addition, mice that were deficient of NGL-2 (*Lrrc4^−/−^* mice) showed excessive repetitive self-grooming and deficits in social behaviours and learning [226]. Although NMDAR-mediated synaptic transmission at SC-CA1 synapses was normal in *Lrrc4^−/−^* mice, both LTP and LTD were significantly suppressed, indicating that a lack of NGL-2 does not affect the synaptic expression of NMDARs, but may interfere with signalling pathways beyond NMDAR activation that are crucial to the expression and maintenance of synaptic plasticity [226].

Other postsynaptic adhesion molecules implicated in ASDs include a family of neuronal postsynaptic cell adhesion molecules called neuroligins [214,227,228,229,230,231]. Neuroligins are expressed differentially in different neuronal types in an isoform-specific manner—neuroligin1 at excitatory synapses [232,233], neuroligin 2 at inhibitory synapses [232,234,235], and neuroligin 3 at both [236]. Furthermore, neuroligins interact trans-synaptically with neurexins, together modulating synapse formation and specification, as well as NMDAR regulation [232,235,237,238,239,240,241]. Genetic alterations in *NLGN1*, which encodes neuroligin 1, have been implicated in human ASDs [230,231,242,243,244]. Neuroligin 1-deficient (Nlgn1^−/−^) mice displayed increased repetitive grooming and impaired spatial memory [245], together with reduced NMDAR-mediated synaptic responses and LTP at SC-CA1 synapses [232,245,246]. In contrast, a mouse model with a human ASD mutation (*Nlgn3^R/C^* mice; the Arg^451^→Cys^451^ substitution in neuroligin-3) exhibited social interaction deficits, and significantly altered NMDAR expression and function in the hippocampus (but not in layer 2/3 of the somatosensory cortex), including abruptly increased protein expression of GluN2B subunits and enhanced NMDA/AMPA ratio and LTP, while mice lacking neuroligin 3 (*Nlgn3^−/−^*) did not display any of these alterations [247], featuring neuroligin isoform/brain region/synapse-specific and genetic variation-dependent changes in NMDAR function in ASD animal models.

Another synaptic cell-adhesion-pathway-related ASD high-risk factor gene is *CNTNAP2*, which encodes the contactin-associated protein 2 (CNTNAP2 or CASPR2) [248,249]. CNTNAP2 is a type I trans-membrane protein, highly homologous to neurexins, that interacts with the post-synaptically localised cell-adhesion membrane protein, contactin 2, and plays an important role in the recruitment of K^+^ channels to the juxtaparanodal regions in myelinated axons [250,251,252], the trafficking of GluA1 subunits of AMPARs in spines [253], and dendritic arborisation and dendritic spine maturation [254]. *Cntnap2^−/−^* mice demonstrated deficits in social communication and interactions, as well as excessive grooming and digging behaviours [255], and exhibited reduced NMDA/AMPA ratio in the hippocampus, although high-frequency stimulated LTP was normal [256]. Lastly, contactin-binding protein 2 (CTTNBP2) is a synapse-localised actin cytoskeleton regulator that controls dendritic spine formation and maintenance [257,258,259], and its mutations have been shown to be linked to ASDs [151,224,260,261]. *Cttnbp2*-deficient (*Cttnbp2^−/−^*) mice demonstrated impaired spatial memory and reduced social interaction, together with reduced protein levels of SHANK3, GluN1, and GluN2A in the synaptosomal fractions [259].

It seems obvious that synaptic proteins encoded by high-risk ASD genes that directly interact with NMDARs and/or play important roles in excitatory synaptic function, including NMDAR signalling, induce functional alterations in NMDARs. However, hundreds of ASD-associated genes do not all simply converge on pathways involved in synaptic transmission, but are also associated with transcription regulation, chromatin modification, and early brain development [152,262,263,264,265]. One example is *TBR1*, a high-confidence ASD risk gene that encodes a T-box transcription factor 1 (TBR1) expressed specifically in the projection neurons of the cerebral cortex, amygdala, and olfactory bulb, which regulates neuronal migration and projection, and cortical lamination [266,267,268,269,270,271]. Interestingly, neuronal activation-dependent expression of the GluN2B subunit of NMDARs is regulated by *Tbr1* through forming a complex with the synaptic protein CASK (calcium/calmodulin-dependent serine protein kinase) and CASK-interacting nucleosome assembly protein (CINAP) [216,272,273]. As expected, *Tbr1*-haploinsufficient mice (*Tbr1^+/−^*) exhibited impaired NMDAR-mediated synaptic response at the thalamic–lateral amygdala synapses [121,195] as well as ASD-like behavioural deficits, such as reduced social interaction, fear memory dysfunction, cognitive inflexibility, and perturbed olfactory discrimination [121,194,268]. These data further imply that NMDAR dysfunction could be one of the common pathophysiological features beyond the synaptic gene alterations identified in ASDs.

### 2.3. NMDARs in Non-Genetic ASD Models

Although ASDs are highly heritable and demonstrate a strong genetic aetiology, nongenetic environmental factors are also considered to significantly contribute to the incidence of ASDs [274,275]. These include prenatal exposure to neurotoxic substances (e.g., pollution, insecticide/pesticide, and toxic heavy metals), low zinc, or disrupted maternal–fetal immune homeostasis (e.g., viral/bacterial infection, autoimmune encephalitis, and maternal autoantibody-related ASD [MAR-ASD]) during a critical period of brain development in the offspring. In particular, MAR-ASD is a subtype of autism in which maternally produced autoantibodies enter the fetal brain, where they induce neurodevelopmental alterations and, thereby, underlie the autistic behaviours in the exposed offspring [276]. Among several autoantibodies identified in MAR-ASD [277,278,279,280], a maternal antibody against NMDA has been suggested [281]. However, another study utilising gestational plasma collected from a subset of Danish participants revealed that autoantibody reactivity to NMDA has a strong correlation to intellectual disability, but not ASDs [282]. Rather, anti-NMDAR encephalitis, an acute neurological disorder caused by autoimmune dysfunctional autoantibodies against NMDARs with yet unknown origin (but perhaps via virus or tumour), has been consistently linked to ASD [283,284,285]. Altogether, despite further investigation being required to confirm the mechanistic underpinning between autoimmune reactivity to NMDAR and ASDs, these findings highlight the significance of NMDARs in normal brain development.

In addition, the implication of maternal infection during pregnancy increasing the risk of ASDs in offspring has been reported [286,287,288]. So far, animal models of ASD-associated maternal infection during gestation (also called maternal immune activation [MIA] models) have been designed by exposing pregnant mothers to synthetic double-stranded RNA, polyinosinic:polycytidylic acid poly(I:C), or bacterial antigen, lipopolysaccharide (LPS), to mimic the viral or bacterial infections, respectively [289,290,291,292,293,294,295,296,297]. The offspring of these MIA animals presented core ASD behaviours, including increased repetitive grooming, heightened anxiety, and impaired social interaction and communication [289,290,291,292,293,294,295,296,297]. Mechanistically, it is thought that MIA-induced dysregulated production of cytokines, such as IL-6, IL-17a, TNFα, and IFNγ, from the mother leads to inflammation in the placenta or directly in the brain of the offspring, which then drives abnormal brain development and, thus, underlies the ASD-linked behavioural deficits [289,290,292,293,294,295,296,297]. Specific to NMDARs in MIA models, there are limited data. Mice with gestational LPS exposure showed a reduction in the protein levels of the GluN2A subunit in the prefrontal cortex, together with significantly decreased dendritic length and spine density [298]. Western blot analysis conducted on the total forebrain lysate of juvenile Wistar rats with prenatal Poly(I:C) treatment showed a decrease in the GluN1 subunit [299], and prenatal Poly(I:C) exposure increased *Grin2a* (encoding the GluN2A subunit) in the prefrontal cortex of Sprague–Dawley rats, as measured by quantitative real-time PCR [300]. It will be of great interest to further experiment on how NMDAR-mediated synaptic transmission and plasticity are altered in MIA models.

Another environmental factor that contributes to the incidence of ASDs is medication during pregnancy [301,302,303]. Prenatal exposure to valproic acid (VPA), an anti-convulsant or anti-epileptic drug, especially in the first trimester of pregnancy, has been recognised as a high-risk factor for ASDs [303], in addition to other side effects, such as neural tube defects, facial abnormalities, developmental delay, and reduced intelligence [304,305,306,307]. The causal relationship between embryonic exposure to VPA and the development of ASD-associated behaviours, such as social deficits and repetitive behaviours, have been well validated in a large number of animal models [212,308,309,310,311,312,313,314,315,316,317,318]. The suggested mechanistic underpinnings of VPA-induced increased risk of ASD include enhanced neural proliferation and neurite growth [319,320]; altered histone acetylation and histone methylation activity [321,322,323,324]; increased neuronal excitability and cortical hyperconnectivity [319,320,325,326,327,328]; and disrupted synaptic transmission and plasticity, including NMDAR signalling [212,326,327,328,329,330,331,332,333]. Prenatal VPA-treated rats on postnatal days 12–16 (P12–16) revealed increased protein levels of GluN2A and GluN2B subunits, larger NMDAR-mediated peak currents and charge transfer, and enhanced LTP in the somatosensory cortex [326]. In addition, the NMDA/AMPA ratio measured from layer 2/3 pyramidal neurons of the medial prefrontal cortex (mPFC) displayed enhanced NMDAR-mediated response during the postnatal period (P8–19), but a normal ratio during adolescence (P22–38) [334]. Intriguingly, NMDAR-mediated synaptic responses (NMDA/AMPA ratio) were reduced and LTP was impaired in the mPFC of mature adult VPA rats (P110–130) [335], indicating an age-dependent switch in NMDAR disruption. Strangely, a suppression of NMDAR function via a blocker, memantine, rescued repetitive behaviours and social deficits in mice of 8–16 weeks old (P56–112) that were prenatally exposed to VPA [212], which presumably indicates NMDAR hyperfunction in these mature adult VPA mice; these findings were opposite to the findings from a rat model of prenatal VPA treatment [334,335]. A recent VPA mice study showed that protein levels of GluN1, GluN2A, and Glun2B subunits were significantly elevated in the somatosensory cortex of 10-week-old (P70–80) VPA mice [330], indicating a potential NMDAR hyperfunction phenotype at least in the somatosensory cortex of adult mice that were prenatally exposed to VPA. These studies imply that a potential species difference or model to examine variability may exist and, thus, needs to be carefully considered.

Prenatal zinc deficiency is an additional non-genetic model for ASDs, with causal links being shown in animal models exhibiting ASD behaviours, learning and memory impairments, and neuropsychological symptoms [336]. SHANK proteins are significantly affected in these mice, with reductions in SHANK1, SHANK2, and SHANK3. As discussed previously, this disruption of developmental synaptic increases in SHANKs subsequently perturbs activity-dependent recruitment of key SHANK binding proteins, including NMDA and AMPA receptor subunits, therefore significantly influencing NMDAR signalling from the early prenatal period [105,195,211,336].

## 3. Effects of Zinc on NMDARs in ASDs

Since disrupted NMDAR signalling plays a crucial role in ASD pathogenesis, it is logical that the restoration of normal NMDAR function has recently been a popular therapeutic mechanism examined in the field. Adult re-expression of *Grin1* by gene editing with Cre recombinase in *Grin1* knockdown mice restored NMDAR responses and synaptic GluN1 protein levels in the medial prefrontal cortex, and rescued social behaviour deficits and improved cognitive function [337]. In addition, pharmacological intervention to improve NMDAR hypofunction, such as D-cycloserine, a partial agonist at the glycine-binding site of NMDARs, has demonstrated therapeutic effects in ASD animal models, including *Shank2^−/−^* [187], *Lrrc4^−/−^* [226], and *Tbr1^+/−^* mice [268]. In agreement with animal data, the administration of D-cycloserine in autistic individuals has shown promising outcomes, including improving social withdrawal and ameliorating repetitive behaviours [338,339]. As NMDAR signalling disruption can be bidirectional in ASDs, the NMDAR antagonists, memantine, agmatine, and MK-801, had also been tested on ASD animals that displayed NMDAR hyperfunction. For example, inhibition of NMDAR hyperfunction in *IRSp53^−/−^* mice normalized social behavioral deficits and neuronal firing in the hippocampus [340]. Treating VPA-exposed rats with agmatine or MK-801 was found to restore excitatory–inhibitory balance and rescue social deficits, as well as repetitive and hyperactive behaviors [310,314]. Furthermore, the rescue of ASD behaviours via treatment strategies, which did not target NMDAR function directly but were aimed at other dysfunctional pathways, such as histone modification and actin cytoskeletal organisation at synapses, was also accompanied with the normalisation of NMDAR function [113,341,342,343,344], further confirming NMDARs as a strong therapeutic target for ASDs. The timing of intervention is critical in modulating NMDAR function, depending on the ASD-associated gene target of interest [207]. Opposite to NMDAR hypofunction observed in adult *Shank2^e6–7−/−^* mice [187], mice at preweaning age (P14) exhibited NMDAR hyperfunction, and early prevention with memantine impeded the development of ASD behaviours, as well as NMDAR hypofunction at later stages [208].

A key ASD therapeutic agent that has shown its effectiveness in treating ASDs through NMDAR modulation is zinc [344,345,346,347]. The hypothesis of zinc supplementation as a potent treatment strategy for ASDs has stemmed from reports that reduced serum zinc levels have been identified in autistic individuals (discussed further in Section 4) [348,349,350,351,352]. To date, a limited number of studies have examined the therapeutic outcome of zinc supplementation in animal models. These include prenatal teratogenic agent-induced animal models of ASD [353,354,355,356], and *Shank2^e6–7−/−^* [171], *Shank3^e13–16−/−^* [161,211], *Tbr1^+/−^* [349,350], and *Cttnbp2^−/−^* mice [357]. Exposure to teratogenic agents, including LPS, urethane, and pro-inflammatory tumor necrosis factor α, in pregnant mothers induces activation of a key zinc-binding protein called metallothionein, which subsequently causes a decrease in plasma zinc levels and ASD-like aberrant behaviours in the offspring [355,358,359,360]. Offspring of mice from dams exposed to LPS to induce maternal inflammation showed perturbed object recognition memory, which was normalised when fed with a zinc-supplemented diet (100 mg Zn/kg) [356]. Moreover, impaired ultrasonic vocalization of rat pups prenatally exposed to LPS was also reduced when zinc treatment was performed 1 h after LPS application in pregnant mothers [355]. Treating prenatally VPA-exposed rats 1 h later with 2 mg/kg of ZnSO_4_ also significantly attenuated abnormal ultrasonic vocalizations, deficits in social interaction and cognitive ability, and repetitive behaviors, even though these behavioural traits were not corrected to those of control [353]. Although promising, these studies did not conduct functional analyses to examine the molecular and cellular mechanisms behind the zinc supplementation-induced rescue of ASD behaviours. Therefore, it would be of great future interest to investigate the role of NMDAR-dependent signalling or synaptic underpinnings of zinc supplementation therapies in animal models of ASD-associated prenatal modification.

In contrast to ASD-associated prenatal rodent models, therapeutic zinc supplementation-dependent changes in NMDAR function have well been explored in transgenic mice deficient with ASD-linked synaptic proteins. Under a regular zinc diet (30 parts per million, ppm), *Shank3^e13–16−/−^* mice of 9–10 weeks old displayed heightened anxiety and repetitive grooming, and impaired social novelty recognition, in comparison to the wild-type mice [161]. However, when *Shank3^e13–16−/−^* mice were fed with a supplementary zinc diet (150 ppm) for six weeks post-weaning, the ASD-associated behavioural deficits were prevented. The amplitude of NMDAR-mediated EPSCs at corticostriatal synapses in *Shank3^e13–16−/−^* mice was intact, but the NMDAR decay kinetics were significantly elevated. Interestingly, the dietary zinc supplementation given to *Shank3^e13–16−/−^* mice not only normalised the NMDAR decay kinetics but also suppressed NMDAR-mediated synaptic response and prevented LTP [161], which is thought to, in part, underlie the supplemented zinc diet-induced normalisation of repetitive self-grooming as corticostriatal synapses contribute to restricted repetitive behaviours [361]. Notably, this dietary zinc supplementation strategy was also effective in the offspring when given maternally during pregnancy and lactation [211]. The supplemented maternal zinc diet (150 ppm) normalised ASD-like behaviours, including social interaction deficits, enhanced anxiety, and increased repetitive self-grooming, in juvenile *Shank3^e13–16−/−^* offspring mice, and, remarkably, these beneficial effects extended into adulthood [211]. In these mice, the reduced NMDAR-mediated EPSC amplitude at corticostriatal synapses was also restored. One thing to note is that even the expression of NMDAR-mediated synaptic deficits at corticostriatal synapses in *Shank3^e13–16−/−^* offspring mice are dependent on maternal zinc intake. The amplitude of NMDAR-mediated EPSC was intact in adult *Shank3^e13–16−/−^* offspring mice when mothers were fed with standard chow with 75 ppm zinc [161], however, it was significantly decreased in adult *Shank3^e13–16−/−^* offspring mice when mothers were given 30 ppm zinc diet [161]. These data indicate that prenatal zinc change can modulate age-dependent changes in NMDAR function. For *Shank2^e6–7−/−^* mice, a slightly different zinc supplementation strategy has been tested. *Shank2^e6–7−/−^* mice acutely treated with clioquinol (30 mg kg^−1^), a zinc chelator, as well as an ionophore [362], intraperitoneally 2 h prior to the experiments demonstrated normalisation of social interaction, as well as normalisation of NMDA/AMPA ratio measured at the hippocampal SC-CA1 synapses in mice of 2–4 months old. This therapeutic effect occurred via the trans-synaptic mobilisation of zinc induced by clioquinol and, at postsynaptic sites, the mobilised zinc induced a potentiation of NMDAR function through the activation of Src tyrosine kinases [87,88,171]. However, acute adult restoration of NMDAR signalling in *Shank2^e6–7−/−^* mice by clioquinol did not rescue deficits in social novelty recognition and elevated anxiety [171], suggesting that certain ASD-associated behaviours cannot be altered in adult animals and/or clioquinol-mediated trans-synaptic zinc mobilisation may not have occurred or be effective in other synapses or brain regions critical for social behaviours and anxiety.

The therapeutic effect of zinc supplementation has been further examined in non-SHANK-related ASD mouse models. Zinc supplementation in *Tbr1^+/−^* mice, either acutely by clioquinol or chronically through a high zinc diet (150 ppm) for 6–8 weeks, rescued social interaction behaviours and normalised NMDA/AMPA ratio at the thalamic-LA synapses, as well as synaptic GluN1 density in the amygdala [171,361]. Since auditory input from the thalamus to the lateral nuclei of the amygdala is crucial for fear memory, the normalisation of glutamatergic thalamic-LA synaptic function (as evidenced by the restoration of AMPAR-mediated synaptic response in addition to NMDAR function) may underlie the prevention of auditory fear conditioning memory deficits in *Tbr1^+/−^* mice [361]. Because thalamic-LA synapses lack free zinc [363], the supplementary zinc diet-induced recovery of the thalamic-LA synaptic function in *Tbr1^+/−^* mice has been attributed to changes in zinc-binding synaptic proteins, such as increased synaptic expression of SHANK3 [361]. However, social novelty recognition defects and impaired axonal projections in the anterior commissure found in *Tbr1^+/−^* mice were not restored by dietary zinc supplementation when given from post-weaning age [361]. In another ASD mouse model, *Cttnbp2^−/−^* mice, daily additional zinc supplementation through drinking water (40 ppm) for seven days improved social deficits and restored the synaptic expression of SHANK2 and SHANK3, as well as GluN1 and GluN2B subunits in the hippocampus [259]. However, one week after the discontinuation of the 7-day zinc supplementation, social interaction behaviours declined and reduced spine density in the hippocampus was not rescued [150]. Altogether, these studies suggest that the timing and duration of zinc-based therapy are crucial to exhibit long-lasting therapeutic effects and not all ASD-associated behaviours are underpinned by zinc-sensitive neural pathways and/or can be rescued by zinc supplementation.

## 4. Therapeutic Potential of Zinc in Humans

Although animal studies have demonstrated the therapeutic potential of zinc, further research is needed to determine whether this potential translates to humans. The hypothesis that zinc supplementation will be beneficial in Phelan McDermid syndrome suggests that elevating zinc drives SHANK3 into synapses and recruits further SHANK2 [344]. This would, in turn, modulate NMDARs and alter excitation/inhibition ratios. However, the dosage and absorption need to be carefully considered and alternate delivery methods, such as nanoparticles for CNS-targeted delivery, will likely be needed [364].

There is some controversy in the literature with regard to whether people with ASDs show zinc deficiency. Several meta-analyses found that patients with ASDs had lower blood levels of zinc, but there was no difference in zinc levels in hair, teeth, or nails [346,365]. Lower levels of zinc in hair samples from children with ASDs have also been reported [354]. In contrast, other studies have reported normal zinc levels in hair, nails, and blood of children with ASDs [366,367]. Furthermore, a clinical trial assessing the levels of zinc and copper in children with ASDs who were 3–8 years old found no significant difference in the levels compared to neurotypical children (https://clinicaltrials.gov/ct2/show/NCT00325572?term=zinc&cond=Autism+Spectrum+Disorder&draw=2&rank=2 accessed on 1 January 2020). This trial was terminated before proceeding to the zinc supplementation phase. Correlations between zinc levels in hair and severity of ASD symptoms have been reported [368,369,370], but it is unclear how zinc measurements in the periphery relate to levels in the brain. It is well known that measurement of zinc levels is challenging, especially of free versus bound zinc, and how blood/hair measurements correlate with zinc level in the brain; together, these likely contribute to the controversy in this field.

A recent small study of zinc supplementation in 30 paediatric subjects with ASDs showed improvements in motor performance as measured using the Childhood Autism Rating Scale (a functional marker of autism severity) after three months of dietary zinc supplementation [371]. This suggests that dietary zinc supplementation could be a viable strategy to ameliorate symptoms in children with ASDs. However, no control group was included in that study. In agreement with a lack of zinc deficiency in ASDs, a meta-analysis including 25 studies and over 11,000 participants found a lack of efficacy of zinc supplementation on child mental and motor development up to nine years of age [372].

Zinc supplementation during pregnancy has also been studied. One study found that zinc supplementation has no effect on pregnancy outcome or birth weight [373]. Similarly, a Cochrane meta-analysis that included 25 randomised controlled trials and over 18,000 pregnant women concluded that zinc supplementation during pregnancy has no detrimental effect on infant outcome [374]. When the development of children born to supplemented or not-supplemented mothers was assessed, there were no differences at 6 and 13 months of age [375]. There is a need for further longitudinal studies that re-assess children when they are older. A clinical trial that will assess the effects of zinc supplementation during pregnancy and lactation in children up to 20 months is underway and the results will be published by 2025 (https://clinicaltrials.gov/ct2/show/NCT04983667?term=zinc&cond=Autism+Spectrum+Disorder&draw=2&rank=1 accessed on 1 January 2020). Together, these studies form an important basis for potential future studies examining zinc supplementation in families with a high risk of ASD, or with prenatal diagnoses; however, clearly more cellular, genetic, and population-based ASD studies are required first.

## 5. Limitations and Future Directions

It is critical to note that the majority of ASD research has been performed in vivo in rodent ASD models, and in rodent cells expressing ASD-causative mutations in vitro. Therefore, although these rodent studies are promising, the influence of zinc supplementation and its glutamatergic synaptic effects need to be investigated in human neurons and, ultimately, in the human brain and in people profoundly affected by ASDs. Information is missing that is critical for the potential translation of zinc-targeted therapies in humans, particularly studies examining how zinc supplementation affects human neuronal or synaptic function. Zinc supplementation also needs to be specifically examined in ASD patients with mutations in key pathways involving *SHANKs* and NMDARs, which animal studies have shown to be a major target for effective zinc-induced therapies. It is clear that a strong platform has been laid by the data from animal studies that support a critical role of zinc in the development of treatment strategies for severe forms of ASDs, such as those with *SHANK* mutations. The next major challenges for the field include identifying key zinc targets for treatment, including NMDARs, and, ultimately, determining the translatability of animal model work into human cells and clinical development.

## Figures and Tables

**Figure 1 pharmaceuticals-16-00001-f001:**
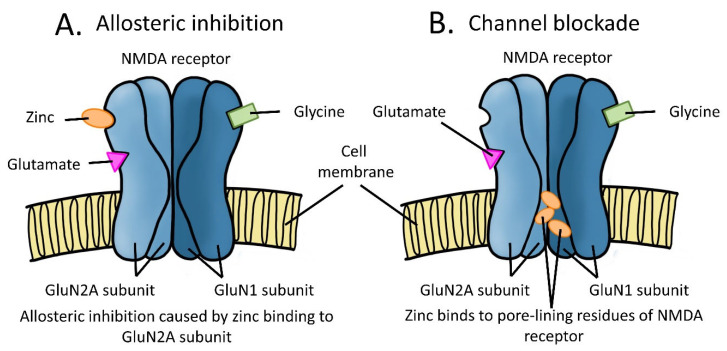
Schematic diagram summarising the inhibitory effects of zinc on NMDARs. (**A**) High-affinity binding of zinc to the GluN2A subunit causes allosteric inhibition by reducing the probability of the channel to open. (**B**) Low-affinity binding of zinc to the residues that line the pore of the NMDAR blocks the channel.

**Figure 2 pharmaceuticals-16-00001-f002:**
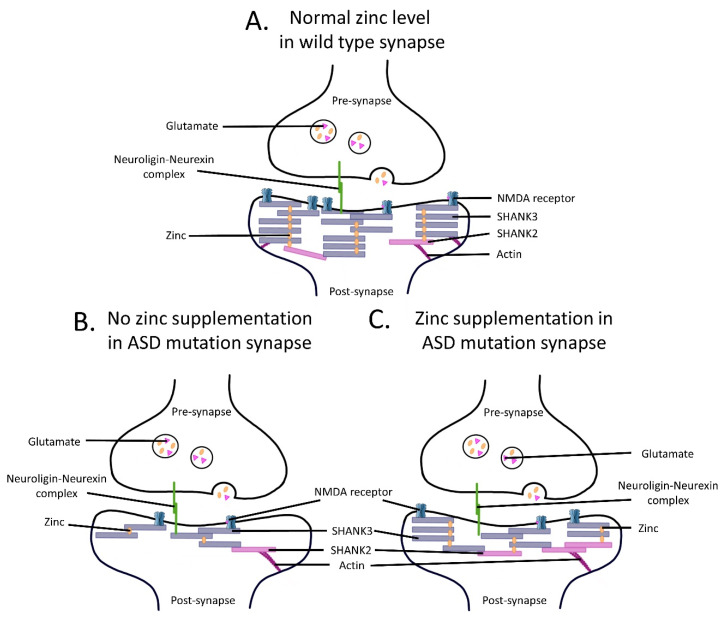
Schematic diagram summarising the positive effects of zinc on glutamatergic synapses through SHANK and recruitment of synaptic proteins, including NMDARs (depicted in blue on the postsynaptic membrane). (**A**) Wild-type synapses are stabilised by the complex organisation of post-synaptic density proteins, including the SHANK scaffold proteins. SHANK proteins link membrane receptors, such as NMDARs, to the actin cytoskeleton. Furthermore, the binding of neuroligin-neurexin complexes to SHANK proteins in the postsynapse enables trans-synaptic signals to the pre-synapse to co-ordinate synaptic plasticity. Zinc maintains the complex organisation of the SHANK proteins in the postsynapse. (**B**) ASD mutation effects on glutamatergic synapses. Depletion of zinc induces the disintegration of SHANK protein complexes, thereby weakening and destabilising the glutamatergic synapse. (**C**) Zinc supplementation can restore the function of the postsynapse by recruiting and stabilising SHANK protein complexes.

## Data Availability

Data sharing not applicable.

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
