# Peer review of "The Role of Zinc and NMDA Receptors in Autism Spectrum Disorders"

_pharmaceuticals, 2022, doi:10.3390/ph16010001_

Round 1
Reviewer 1 Report
I read with interest the article "The role of zinc and NMDA receptors in Autism Spectrum Disorders" by Lee and co-authors, which reviewed the research on ionotropic NMDA and AMPA receptors, some structural synaptic proteins (SHANK1, 2, 3, IRS, etc.) in relation to the functioning of synapses and synaptic plasticity in different models of autism spectrum disorders compared with the normophysiological conditions. The review includes relevant and recent research in this area, is logically structured and is read with interest. I have no fundamental remarks, the relevance of the topic and the exhaustive presentation of the data are beyond doubt.
Author Response
We thank Reviewer 1 for their comments on our manuscript and note there are no further changes required.
Reviewer 2 Report
1) Please unify the font size and style in the introduction section.
2) In Section 1.1 state the brain regions expressing the densities of NMDAR subunits mentioned
3) In Section 1.2 there is controversial findings in line 102 and 103 that requires further elaboration
and references of more recent articles are recommended.
4) From line 121 the paragraph needs to be rephrased and referenced for better understanding
5) Figure1 and Figure 2 need to be more demonstrative. In Figure 2 mention the type of receptors expressed postsynaptically.
6) Line 192: Rephrase this sentence into an informative statement rather than a question format.
7) Further evidence needs to be mentioned elaborating the role of Zinc and SHANK with APMAR, NMDAR and heterodimer colocalization of AMDAR/NMDAR.
8) Line 195: Mention all brain regions implicated.
9) Line 206: remove A) and B). A continuous sentence is preferred.
10) Line 211: Please rephrase this sentence.
11) Line 223: Remove the brackets before the reference.
12) For section 2.1 I suggest adding a table including all ASD associated shank animal models, specific domains, effect of the gene of defect and the phenotype of the animal model.
13) I suggest adding another table summarizing the information in 2.2 and 2.3.
14) Line 368: change the reference style to the style used in all references of the manuscript.
15) Line 416: Remove the bracket before the reference.
16) Line 458: GRIN2A instead of Grin2a.
17) Line 493: Please rephrase the paragraph with more scientific terms
18) Line 495: GRIN1 instead of Grin1
19) Line 507: What are VPA rats. Please clarify
20) Line 514: No need for ‘One thing to note is that’
21) Line 515: Add space before opposite.
22) Line 516: ‘these mice’ refers to which mice, please make it clear in the sentence.
23) Line 562: The sentence is too long.
24) Line 577: Break down the sentence to make it more meaningful.
25) Line 619 and 644: Use the reference used in the whole manuscript.
26) Line 658: Rephrase and break the sentence.
27) All molecular effect of zinc in ASD animal models and humans must be included in this review.
28) Please check in text referencing of reference 297 (not available in text)
29) All the bibliography need to be revised with the in text referencing.
Author Response
We thank Reviewer 2 for their input and below we have detailed the changes we have made in the manuscript to address these.
1) Please unify the font size and style in the introduction section.
All text is now formatted Calibri, font size 11.
2) In Section 1.1 state the brain regions expressing the densities of NMDAR subunits mentioned.
We have now included additional text detailing the relevant brain regions in which NMDAR subunit expression was studied in this section.
3) In Section 1.2 there is controversial findings in line 102 and 103 that requires further elaboration and references of more recent articles are recommended.
We have now included additional text and references to expand on the discussion point on NMDAR regulation during LTP.
4) From line 121 the paragraph needs to be rephrased and referenced for better understanding.
We have updated this paragraph to clarify the focus on how the state of a synapse defines its ability to undergo plasticity, and included references for this work.
5) Figure1 and Figure 2 need to be more demonstrative. In Figure 2 mention the type of receptors expressed postsynaptically.
We have included additional detail into both figures and added text to the figure legend with information on postsynaptic receptors.
6) Line 192: Rephrase this sentence into an informative statement rather than a question format.
This sentence is rephrased as a statement.
7) Further evidence needs to be mentioned elaborating the role of Zinc and SHANK with APMAR, NMDAR and heterodimer colocalization of AMDAR/NMDAR.
We have included detailed discussion on the role of zinc in recruiting SHANK to synapses and the subsequent increase in postsynaptic receptor localisation, summarising work from the references [89, 97, 98, 154, 164, 204, 339-342, 357-361]. We have now expanded on the text describing this work in Section 1.3. Our text remains concentrated on NMDARs as these receptors are the focus of this review.
8) Line 195: Mention all brain regions implicated.
We have added details of the hippocampal studies in the research described.
9) Line 206: remove A) and B). A continuous sentence is preferred.
We have updated this sentence to list 1) and 2) as the sentence is about the two key criteria for ASD diagnosis
10) Line 211: Please rephrase this sentence.
This sentence has been re-ordered and shortened.
11) Line 223: Remove the brackets before the reference.
Bracket has been removed
12) For section 2.1 I suggest adding a table including all ASD associated shank animal models, specific domains, effect of the gene of defect and the phenotype of the animal model.
We have recently generated this table in the 2021 published review: Vyas Y, Cheyne JE, Lee K, Jung Y, Cheung PY, Montgomery JM. Shankopathies in the Developing Brain in Autism Spectrum Disorders. Front Neurosci. 2021 Dec 22;15:775431. doi: 10.3389/fnins.2021.775431. We now refer directly to this table in the manuscript.
13) I suggest adding another table summarizing the information in 2.2 and 2.3.
We are not convinced that this would be helpful. Sections 2.2 and 2.3 summarise a large amount of recently published literature that could not be condensed into a table.
14) Line 368: change the reference style to the style used in all references of the manuscript.
This has been corrected.
15) Line 416: Remove the bracket before the reference.
Bracket removed.
16) Line 458: GRIN2A instead of Grin2a.
This text refers to rodent gene work so is Grin2a.
17) Line 493: Please rephrase the paragraph with more scientific terms
It is not clear which terms are required. Can the reviewer please specify?
18) Line 495: GRIN1 instead of Grin1
This text refers to rodent gene work so is Grin1.
19) Line 507: What are VPA rats. Please clarify
VPA is introduced and described in Section 2.3, paragraph 3.
20) Line 514: No need for ‘One thing to note is that’
This has been deleted.
21) Line 515: Add space before opposite.
This has been added.
22) Line 516: ‘these mice’ refers to which mice, please make it clear in the sentence.
We have detailed P14 mice to clarify this sentence refers to younger mice.
23) Line 562: The sentence is too long.
The sentence has been broken down into 3 sentences.
24) Line 577: Break down the sentence to make it more meaningful.
We have bracketed the AMPAR and NMDAR evidence to make the impact of the consequences of the research clear.
25) Line 619 and 644: Use the reference used in the whole manuscript.
These are links for the reader to directly access with the text.
26) Line 658: Rephrase and break the sentence.
The sentence has been broken into two parts.
27) All molecular effect of zinc in ASD animal models and humans must be included in this review.
Our review provides a comprehensive overview focussing on all major studies in ASD mouse models and zinc. But overall the focus is NMDARs in ASD and zinc. If there are further specific studies the reviewer would like to see included please specify these.
28) Please check in text referencing of reference 297 (not available in text)
Reference 297 is referenced in the text in Section 2.3 paragraph 3 in regards to exposure to VPA during pregnancy.
29) All the bibliography need to be revised with the in text referencing.
We have now updated the in text referencing with reference numbers placed in square brackets and placed before punctuation.

Round 2
Reviewer 2 Report
1) Line113: The word ‘however’ is repeated twice. Please delete one without affecting the meaning of the sentence.
2) Line129: Replace the semicolon (:) with full stop (.).
3) Line 484: Please replace ‘VPA rats’ with ‘VPA-exposed rats.’
4) Regarding to my earlier comment of mentioning molecular effect of zinc in ASD animal models and humans, I would prefer if you would consider prenatal zinc deficient mice (PZD) as model for autism spectrum disorder, including the mechanisms involved in the development of the ASD-like phenotype in PZD mice, which further highlights the association of ASD-phenotype and zinc. Refer to the below paper link address: https://doi.org/10.3390%2Fijms23116082
Author Response
- Line113: The word ‘however’ is repeated twice. Please delete one without affecting the meaning of the sentence.
Corrected
- Line129: Replace the semicolon (:) with full stop (.).
Corrected
- Line 484: Please replace ‘VPA rats’ with ‘VPA-exposed rats.’
Corrected
- Regarding to my earlier comment of mentioning molecular effect of zinc in ASD animal models and humans, I would prefer if you would consider prenatal zinc deficient mice (PZD) as model for autism spectrum disorder, including the mechanisms involved in the development of the ASD-like phenotype in PZD mice, which further highlights the association of ASD-phenotype and zinc. Refer to the below paper link address: https://doi.org/10.3390%2Fijms23116082
We have now added an additional paragraph at the end of Section 2.3 NMDARs in non-genetic ASD models to highlight prenatal deficient mice as a model of ASD and the relevant work on this model on NMDARs.